# Across the Myeloablative Spectrum: Hematopoietic Cell Transplant Conditioning Regimens for Pediatric Patients with Sickle Cell Disease

**DOI:** 10.3390/jcm11133856

**Published:** 2022-07-03

**Authors:** Emily Limerick, Allistair Abraham

**Affiliations:** 1Cellular and Molecular Therapeutics Branch, National Health, Lung, and Blood Institute, Bethesda, MD 20892, USA; 2Division of Blood and Marrow Transplantation, Children’s National Hospital, Washington, DC 20010, USA; aabraham@childrensnational.org

**Keywords:** sickle cell disease, allogeneic transplant, conditioning intensity, reduced intensity conditioning, myeloablative conditioning, non-myeloablative conditioning

## Abstract

One out of every five hundred African American children in the United States has sickle cell disease (SCD). While multiple disease-modifying therapies are available, hematopoietic cell transplantation (HCT) remains the only curative option for children with SCD. HLA-matched sibling HCT has demonstrated excellent efficacy, but its availability remains limited; alternative donor strategies are increasingly explored. While Busulfan-Cyclophosphamide has become the most widespread conditioning regimen employed in HCT for pediatric SCD, many other regimens have been examined. This review explores different conditioning regimens across the intensity spectrum: from myeloablative to non-myeloablative. We describe survival and organ function outcomes in pediatric SCD patients who have received HCT and discuss the strengths and weaknesses of the various conditioning intensities. Finally, we posit novel directions in allogeneic HCT for SCD.

## 1. Introduction

Sickle cell disease (SCD) affects over 100,000 people in the US alone. The sequelae may include painful vaso-occlusive crises, chronic pain, acute chest episodes, chronic kidney disease, and cardiomyopathy. SCD is associated with early mortality as well as significant direct and indirect economic costs [1,2,3,4,5]. Improvements in supportive care and disease-modifying therapy have provided new treatment opportunities, but hematopoietic cell transplant (HCT) remains the only potentially curative strategy. As the use of HCT expands, conditioning regimens to meet the needs of SCD patients across the age and morbidity spectrum are necessary. Here, we focus on conditioning strategies employed in pediatric SCD patients.

Operational definitions of conditioning intensity are based on the expected duration of pancytopenia and the need for stem cell support for hematopoietic recovery. The intensity of conditioning regimens can vary significantly. While the conventional conditioning for most young patients with SCD is myeloablative, attempts have been made to limit early transplant toxicity by modifying the intensity of the conditioning regimen. Myeloablative conditioning (MAC) refers to administering total body irradiation (TBI) and/or alkylating agents at doses that will not allow autologous hematologic recovery. While complete myeloablation is likely impossible [6], a practical definition is a combination of agents expected to produce profound pancytopenia and myeloablation within 1–3 weeks of administration; pancytopenia is long-lasting, usually irreversible, and in most cases fatal unless hematopoiesis is restored by hematopoietic stem cell infusion [7].

Non-myeloablative (NMA) conditioning expands HCT availability to older populations and those with significant co-morbidities by minimizing cytopenia and early toxicity through immunoablation; the combination of substantial immunosuppression and larger CD34+ stem cell doses allows donor engraftment [8]. An NMA regimen can therefore be defined as a regimen that causes minimal cytopenia and does not require stem cell support for marrow recovery [7]. Reduced intensity conditioning (RIC) is an intermediate category that differs from MAC and NMA. RIC regimens are associated with cytopenia and require stem cell support. Still, TBI and alkylating agent doses are reduced by at least 30%, so these regimens are defined only by their inability to meet the MAC or NMA definition. For some conditioning regimens, the classification may not be straightforward [7]. For example, some authors believe non-myeloablative regimens may include alkylating agents at a low dose, while others classify such regimens as either myeloablative or reduced intensity. Here, we review myeloablative, non-myeloablative, and reduced intensity conditioning regimens employed in the pediatric SCD setting. Table 1 is adapted from CIBMTR conference proceedings [9] and outlines the conditioning intensity of the regimens most commonly used in this patient population.

## 2. Myeloablative Conditioning Regimens

The first HCT in a patient with SCD was for the treatment of acute myelogenous leukemia in a pediatric patient [10]. Her myeloablative conditioning, which included cyclophosphamide and TBI, was given for her malignancy, but the matched sibling donor transplant demonstrated the potential for cure of SCD. Subsequently, five children with severe SCD received HLA-matched sibling transplants after conditioning with busulfan and cyclophosphamide with complete cessation of vaso-occlusive episodes and hemolysis [11]. Since those first reported HCT cases, busulfan (BU) and cyclophosphamide (Cy) have been the mainstay of conditioning for HLA-matched HCT for pediatric SCD [12,13,14]. Indeed, a 2017 CIBMTR, European Group for Blood and Marrow Transplantation (EBMT), and Eurocord database investigation reported that 87% of MAC conditioning for pediatric patients was with this regimen [15].

Outcomes for pediatric regimens employing a Bu/Cy conditioning have been excellent: overall survival (OS) rates range from 90 to 100% and event-free survival (EFS) from 77 to 100% [13,16,17,18,19,20,21,22,23,24,25]. On the other hand, the same reports describe grade II-IV aGVHD in 11–39% of patients. The BU/Cy regimen is associated with cGVHD rates ranging from 0 to 21%, though GVHD prevention strategies vary [13,16,17,18,19,20,21,22,23,24,25]. 

Despite its demonstrated efficacy, this regimen causes significant toxicity, including invasive infections, severe pancytopenia requiring aggressive transfusion support, mucositis, and hepatic sinusoidal obstructive syndrome [26]. Additionally, infertility is a potential risk of BU/Cy, and surveys have indicated that it is a significant concern for patients and families considering HCT [27,28]. Indeed, of the various classes of chemotherapy agents, alkylating agents, such as busulfan, pose the highest risk to fertility and are associated with premature ovarian insufficiency [29]. Further, alkylators are also associated with therapy-related malignancies, specifically acute leukemia and myelodysplasia [30], as well as delayed lung toxicity; BU, in particular, is known for its potential pulmonary effects [31].

BU/Cy dosing strategies have varied to mitigate drug risks and benefits. Early strategies included BU and Cy at different doses, including 200 mg/kg or 260 mg/kg Cy and 14, 16, or >16 mg/kg of BU [32]. Higher BU doses are associated with hepatic sinusoidal obstructive syndrome, seizures, interstitial pneumonitis, and mortality; lower BU is associated with higher rates of graft rejection [33]. Children treated with high-dose BU achieve lower plasma concentrations, measured by area under the plasma–concentration time curve (AUC) or steady-state concentration (C_ss_), than adults [34]. (The C_ss_ is a counterpart of AUC and is calculated as the AUC divided by dose frequency.) SCD, on the other hand, does not independently impact BU clearance compared to children without SCD [35]. Targeted busulfan dosing to achieve a total C_ss_ of 600–700 ng/mL (14.4–16.8 mg × h/L AUC per dose) has been associated with robust and sustained engraftment in pediatric HCT for SCD. Further, targeting AUC > 900 µMol × min (C_ss_ > 615 ng/mL; >14.8 mg × h/L AUC per dose) may contribute to sustained engraftment [24,36,37]. Although better outcomes have been reported with targeted BU therapy, the use of a BU-based conditioning regimen is established only in the setting of MSD transplant. Less than 20% of SCD patients needing an HCT have a matched sibling available. Therefore, alternative donor approaches are critical [38,39].

Studies in leukemia and thalassemia have demonstrated that substituting fludarabine for cyclophosphamide in MAC improves safety and retains efficacy [40,41]. Therefore, to establish less toxic conditioning strategies in SCD conditioning, fludarabine (flu) has been added to BU/cy to reduce busulfan and cyclophosphamide doses [20]; it has also been proposed as an alternative to cyclophosphamide [42,43,44,45]. Fludarabine is highly immunosuppressive and relatively safe [45]. BU/Flu has been employed in 7.5% of pediatric SCD MAC conditioning regimens [15]. BU/Flu has been employed in the HLA-matched setting (MSD and MUD) [42,44,45], as well as haploidentical HCT [43]. Outcomes of Bu/Flu conditioning vary. While overall survival exceeds 90%, Krishnamurti et al.’s multicenter pilot study of adolescents and young adults reports 60% 3-year EFS among MUD transplants compared to 88% for MSD [42]. Exclusively pediatric cohorts have reported 2-year OS and EFS of 100% with the BU/Flu regimen in the MSD setting [45] and 75% EFS in the MUD context [44]. Pawlowska describes early results in a small cohort of adolescents and young adults; they achieved 100% OS and EFS with up to 11 months of follow-up [43]. GVHD rates are similarly variable: no acute or chronic GVHD was reported in the MUD patients. While no haplo patients developed aGVHD, 75% developed cGVHD, primarily mild and limited. aGVHD was noted in 18% of MSD and 27% developed cGVHD [42,43,44,45].

Treosulfan (treo) is a structural analog of BU with significant myeloablative potential, immunosuppressive activity, and minimal extramedullary effects [46]. It has been employed in HCT for malignant and non-malignant diseases with low toxicity and demonstrated efficacy [47,48]. Therefore, another attempt to reduce toxicity in patients with SCD within the myeloablative framework includes conditioning with flu, thiotepa (thio), and treo [19,49,50,51,52]. An analysis of unrelated donor HCT for SCD reported that 3-year OS in patients treated with flu/thio/treo conditioning was higher than in patients treated with other regimens. However, the 3-year GVHD-free, relapse-free survival (GRFS) was not significantly different [19]. Others have reported excellent outcomes even in the alternative donor setting with 100% primary engraftment and OS and EFS 90% and higher [49,50,52]. Of note, Marzollo and colleagues describe a cohort of 11 children transplanted after flu/thio/treo. Donor sources included MSD, haplo, MUD, and MMUD. All patients engrafted, and there were no instances of secondary graft failure. They detail stable mixed chimerism in a large proportion of patients [50]. While treo has been approved for use in Canada and Europe, the FDA has not yet approved its use in the USA [53].

## 3. Reduced Intensity and Non-Myeloablative Conditioning Regimens

Despite these efforts to reduce toxicity, MAC remains a significant risk and is often prohibitively damaging, particularly in older adults—registry data have demonstrated that 5-year survival is 80% in patients >16 years compared to 95% in those <16 years—and those with significant comorbidities [15]. Thus, reduced intensity and non-myeloablative conditioning regimens have been employed with various transplant types, including matched sibling and matched unrelated donor, mismatch unrelated, and haploidentical HCT. Understanding that complete donor chimerism is unnecessary for SCD symptoms to resolve has allowed expansion of alternative donor strategies coupled with less intense conditioning regimens. Mixed chimerism occurs after allogeneic HCT when the lymphohematopoietic system comprises both donor and recipient-derived blood cells [54]. Stable mixed chimerism was also described in Strocchio’s analysis; they hypothesize that the tolerance achieved in these patients may indeed also be associated with lower GVHD rates [51], a concept previously demonstrated in thalassemia patients [55]. Younger age, use of ATG, and cord blood vs. bone marrow sources have been associated with mixed chimerism [56]. Minimum donor myeloid chimerism of 20–25% has been reported to reverse the SCD phenotype [57,58,59]. As alternative donor approaches are necessary to improve the less than 10% availability of matched sibling transplants, the role of RIC and NMA conditioning has increased [60].

### 3.1. Reduced Intensity Conditioning

Fludarabine has been employed as part of reduced intensity conditioning regimens with varying success [17,61,62,63,64,65,66,67]. Two of three haploidentical patients conditioned with an RIC regimen of fludarabine, thiotepa, and busulfan with anti-thymocyte globulin (ATG) and muromonomab-CD3 experienced rejection [17]. A small cohort of children with SCD received a mismatch unrelated donor transplant after RIC with flu, melphalan (mel), alem, and an infusion of marrow-derived mesenchymal stem cells to facilitate engraftment; all patients either died or had autologous recovery [67]. The Sickle Cell Unrelated Donor Transplant (SCURT) trial enrolled 29 patients conditioned with flu, mel, alem [65]. Although the 1-year EFS was 76%, high rates of GVHD and GVHD-related deaths marred the study. Further, the cord blood arm of SCURT stopped early due to high rates of graft rejection [62]. Subsequent work, however, has demonstrated that adding hydroxyurea and thiotepa to the fludarabine/melphalan regimen can improve engraftment after cord blood transplants [61]. American Society of Hematology (ASH) guidelines for stem cell transplantation in SCD recommend using myeloablative conditioning over RIC with flu/mel in pediatric patients with a matched sibling donor available [68].

A small pediatric cohort transplanted with a reduced intensity bu/flu/ATG/total lymphocyte irradiation regimen experienced 86% EFS [69]. Similar RIC regimens in the MSD, unrelated donor and haplo settings have been associated with >90% overall and event-free survival [63,64,66,70]. The ideal RIC regimen that maximizes engraftment and minimizes toxicity is unknown [71].

### 3.2. Non-Myeloablative Conditioning

The NIH platform is a non-myeloablative regimen used in MSD HCT and is aimed at tolerance induction; it includes low-dose TBI and alemtuzumab and has demonstrated efficacy in children and adults [8,72,73,74]. Early non-myeloablative conditioning strategies in the haploidentical setting employed fludarabine and low-dose TBI with meager results: most patients’ engraftment was only transient [75,76]. The Sickle transplant Using a Nonmyeloablative approach (SUN) multicenter clinical trial (NCT03587272) employs the NIH regimen and is ongoing [77]. Interim results report EFS of 87.5% and 100% OS. Likewise, Johns Hopkins researchers have described their experience in young adults with non-myeloablative haploidentical HCT using ATG, flu, cy, TBI, and subsequently an RIC version that added thiotepa [78]. Their cohort, whose median age was 22, experienced 100% OS and 93% EFS. The excellent results of these low-intensity approaches prompted Nickel et al. to suggest reserving myeloablation for a second transplant in the minority of patients who experience graft failure [77]. The ASH SCD stem cell transplant guidelines suggest NMA conditioning over RIC, specifically flu/mel-containing regimens, for adults with SCD who have a transplant indication [79].

### 3.3. T-Cell Depletion Strategies

Authors have hypothesized that exposure to minor histocompatibility antigens through multiple blood transfusions may increase the likelihood of immunologic rejection in SCD patients. ATG, Alemtuzumab, and other T-cell depletion strategies have been employed to mitigate that risk [14,76]. Indeed, in 87 young SCD patients who received an MAC regimen, the addition of ATG was associated with a reduction of the rejection rate from 23% to 3% [13]. The timing of t-cell depletion in an RIC regimen, early vs. late, may impact engraftment. Specifically, ‘early’ alemtuzumab may be associated with improved engraftment, though may carry the risk of increased acute and chronic GVHD [80].

## 4. NMA vs. RIC vs. MAC Pro/Cons

Direct comparison of the outcomes of different conditioning regimens is limited. A transplant database analysis reported that the OS and EFS were highest after NMA conditioning with no significant differences between MAC and RIC [81]. Acute and chronic GVHD rates were higher with MAC and RIC than NMA regimens. However, in analyses limited to MSD transplants, though some have reported that graft failure was higher after RIC [81], multiple reports conclude that conditioning regimen intensity is not associated with overall survival [15,81]. Mixed chimerism (MC) has been reported after NMA, RIC, and MAC, though the incidence varies with different MC definitions. Although it is more common with RIC and NMA preparations, a considerable number (29%) of patients with MC have been described after MAC as well [82]. CD34+ stem cell boosts have been successfully employed in the setting of falling chimerism and return of SCD after RIC and NMA conditioning [83,84]. Finally, although the risk of developing a malignancy post-transplant was seven times higher with NMA versus RIC, this did not reach statistical significance; there were no cases of malignant neoplasm after myeloablative conditioning [81]. 

Additional limitations of direct comparisons include differences in follow-up duration. For example, the median follow-up duration for the predominantly MAC pediatric MSD studies described in a 2017 international registry study was 56.4 months [15]. A 2019 registry study describes median follow-up of surviving patients: MSD, whose predominant conditioning was myeloablative, had 36 months of follow-up, while haploidentical, the majority of whom had NMA conditioning, had 25 months of follow-up [81]. Further, age is a significant variable that differs between donor type and conditioning regimen. MSD patients, for whom MAC predominates, tend to be younger than the predominantly NMA haploidentical recipients [81]. Data have shown that a younger age of transplant is associated with improved outcomes [15]. Table 2 summarizes the available data comparing relative outcomes of different regimen intensities.

## 5. Symptomatic/Phenotypic Improvements after HCT

While all regimen intensities have been associated with reversal of the SCD phenotype and resolution of SCD symptoms [17,18,85,86,87], their late effects on other organ systems are of tremendous import.

***Reproductive***: Most girls require hormone replacement after myeloablative HCT [16]. Girls who were significantly younger at the time of transplant were more likely to have spontaneous puberty. Of 20 women >25 years old, 4 women had six spontaneous pregnancies; they were all transplanted at 5–7 years of age [16]. Others have also reported spontaneous puberty and pregnancies in some girls after myeloablative conditioning with busulfan, including a healthy pregnancy after spontaneous resolution of post-HCT primary ovarian insufficiency [18]. Fertility preservation may be offered before HCT, but potential risks include painful crises and even respiratory failure [88,89]. Mishkin et al. also argue that the psychiatric burden of SCD, HCT toxicities, and resultant infertility may be cumulative [90]. Reproductive system effects have been reported after RIC as well. Three adolescent males all developed oligospermia or azoospermia after RIC [91]. Even in the RIC setting, amenorrhea is noted in patients who are post-pubertal at the time of transplant and spontaneous menarche in patients who are younger in age at the time of transplant [69]. Spontaneous pregnancies have also been reported in men and women after non-myeloablative HCT for SCD in adolescents and adults [8].

***Spleen and kidney***: Improvement in splenic function has been described with the disappearance of Howell–Jolly bodies post-HCT [13]. Stable renal function after NMA HCT has also been reported [8,92].

## 6. Toward the Future

Excellent results have been achieved with matched sibling donor transplants. Clinical trials are ongoing to determine the best approaches to optimize efficacy and minimize toxicity, especially using alternative donor strategies, which are necessary to expand transplant availability. Pre-transplant immunosuppression has been employed to suppress endogenous hematopoiesis, overcome the engraftment barrier, and improve graft failure rates in the haploidentical setting [43,93]. Strategies include pentostatin and cyclophosphamide (NCT# 03077542), dexamethasone and fludarabine [43], and hydroxyurea and azathioprine [94]. Granulocyte stimulants like gCSF are also being employed safely in the post-HCT to enhance neutrophil recovery [95]. Finally, new conditioning strategies that avoid genotoxicity altogether represent the next wave of HCT development. Anti-CD117 and anti-CD45 antibodies have been conjugated with immunotoxins and deployed in experimental models to effectively condition mice without concomitant cytopenias [96,97]. Similarly, chimeric antigen receptor t-cells targeted against CD117 have also been employed in murine models to condition effectively without chemotherapy or radiation [98].

## 7. Conclusions

Myeloablation with Bu-Cy has established efficacy and has become the standard of care in the pediatric MSD setting, but at the cost of significant toxicity and limited tolerability for only younger patients with fewer comorbidities. Reduced intensity and nonmyeloablative strategies may rival myeloablative ones in terms of event-free and overall survival, though clinical trials to assess the effectiveness of specific regimens and conditioning intensity are required. In the meantime, choosing a regimen based on the available donor strategy—particularly for SCD patients requiring an alternative option—demonstrated outcomes, and anticipated toxicity, typically in the clinical trial setting.

## Figures and Tables

**Table 1 jcm-11-03856-t001:** Common transplant conditioning regimens.

Conditioning Regimen
**Myeloablative**
Busulfan (BU) > 7.2 mg/kg IV (or >9.0 mg/kg PO)
BU > 300 mg/m^2^ IV (or 375 mg/m^2^ PO)
Melphalan (Mel) > 150 mg/m^2^
Thiotepa (TT) > 10 mg/kg
Treosulfan (treo) > 30,000 mg/^2^ (or >30 g/m^2^)
**Reduced Intensity and Non-myeloablative**
Total Body Irradiation (TBI) ≤ 5 Gy (single) or TBI ≤8 Gy (fractionated)
Cyclophosphamide +/− Anti-Thymocyte Globulin (ATG) +/− Fludarabine (Flu)
BU ≤ 7.2 mg/kg IV or ≤9.0 mg/kg PO
BU ≤ 300 mg/m^2^ IV or ≤375 mg/m^2^ PO
Mel ≤ 150 mg/m2 +/− Flu
Treo ≤ 30 g/m^2^ +/− Flu
TT ≤ 10 mg/kg

PO per os (by mouth).

**Table 2 jcm-11-03856-t002:** Relative risk of outcomes after transplant by conditioning intensity.

	Myeloablative	Reduced Intensity	Non-Myeloablative
**Overall survival ***	🡅	🡅	🡅🡅
**Event-Free Survival** (= alive without graft failure)	🡅	🡅	🡅
**aGVHD**	🡅🡅	🡅🡅	🡅
**cGVHD**	🡅🡅	🡅🡅	🡅
**Post-transplant malignancy ^**	none reported	🡅	🡅🡅

* In analyses limited to matched sibling donor setting, no association found between overall survival and regimen intensity. ^ difference not statistically significant.

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
