# Peer review of "Across the Myeloablative Spectrum: Hematopoietic Cell Transplant Conditioning Regimens for Pediatric Patients with Sickle Cell Disease"

_jcm, 2022, doi:10.3390/jcm11133856_

Round 1
Reviewer 1 Report
Dear Authors,
Congratulations for this review on different conditionning regimen in HSCT for SCD.
If MAC, RIC and NMA are well described with many references, the pro/cons for one or an other regimen (chapter 4, line 196) should be more developped since this is probably the added value of this paper.
If indeed direct comparisons are limited in the literature, the conclusion of a single paper should be nuanced and consider also confounding factors such as follow-up duration, the longterm outcome of patient with mixed chimerism, the increased risk of malignancy with TBI, age of the patient, ...
Author Response
If MAC, RIC and NMA are well described with many references, the pros/cons for one or another regimen (chapter 4, line 196) should be more developed since this is probably the added value of this paper.
If indeed direct comparisons are limited in the literature, the conclusion of a single paper should be nuanced and consider also confounding factors such as follow-up duration, the longterm outcome of patient with mixed chimerism, the increased risk of malignancy with TBI, age of the patient, ...
Response: Thank you for this suggestion. We have expanded on our discussion to include the following comments:
“Mixed chimerism (MC) has been reported after NMA, RIC, and MAC though the incidence varies with different MC definitions. Although it is more common with RIC and NMA preparations, a considerable number (29%) of patients with MC have been described after MAC as well. CD34+ stem cell boosts have been successfully employed in the setting of falling chimerism and return of SCD after RIC and NMA conditioning”
“Additional limitations of direct comparisons include differences in follow-up duration. For example, the median follow-up duration for the predominantly MAC pediatric MSD studies described in a 2017 international registry study was 56.4 months. A 2019 registry study describes median follow-up of surviving patients: MSD, whose predominant conditioning was myeloablative, had 36months of follow-up while haploidentical, the majority of whom had NMA conditioning, had 25months of follow-up. Further, age is a significant variable that differs between donor type and conditioning regimen. MSD patients, for whom MAC predominates, tend to be younger than the predominantly NMA haploidentical recipients. Data have shown that younger age of transplant is associated with improved outcomes.”
Reviewer 2 Report
Well-written, concise, and timely review on conditioning regimens for hematopoietic cell transplantation for sickle cell disease.
Comments
- Replace or at least add harmonized Busulfan units (mg X h/L) to Css
- While it is clear what constitutes a MAC regimen, the distinction between reduced intensity (RIC) and non-myeloablative (NMA) is often not clear both in Table 1 (where the agents are grouped together) and throughout the manuscript. For example, why is the Johns Hopkins backbone, which most publications consider a RIC, classified here as NMA even after the addition of 10 mg/kg of TT. This is important as Table 2 illustrates potential differences in outcomes between RIC and NMA conditioning.
- In Table 1 Melphalan is misspelled
Author Response
1. Replace or at least add harmonized Busulfan units (mg X h/L) to Css
Response: Thank you; we have added the harmonized Busulfan units for clarity
2. While it is clear what constitutes a MAC regimen, the distinction between reduced intensity (RIC) and non-myeloablative (NMA) is often not clear both in Table 1 (where the agents are grouped together) and throughout the manuscript. For example, why is the Johns Hopkins backbone, which most publications consider a RIC, classified here as NMA even after the addition of 10 mg/kg of TT. This is important as Table 2 illustrates potential differences in outcomes between RIC and NMA conditioning.
Response: In agreement, we have clarified with the following comments:
“For some conditioning regimens, the classification may not be straightforward. For example, some authors believe non-myeloablative regimens may include alkylating agents at a low dose while others would classify such regimens as either myeloablative or reduced intensity.”
“Likewise, Johns Hopkins researchers have described their experience in young adults with non-myeloablative haploidentical HCT using ATG, flu, cy, TBI, and subsequently a RIC version that added thiotepa.”
3. In Table 1 Melphalan is misspelled
Response: thank you, we corrected the spelling